# Anaerobic Performance after a Low-Carbohydrate Diet (LCD) Followed by 7 Days of Carbohydrate Loading in Male Basketball Players

**DOI:** 10.3390/nu11040778

**Published:** 2019-04-04

**Authors:** Małgorzata Magdalena Michalczyk, Jakub Chycki, Adam Zajac, Adam Maszczyk, Grzegorz Zydek, Józef Langfort

**Affiliations:** 1Department of Sports Nutrition, The Jerzy Kukuczka Academy of Physical Education in Katowice, Mikolowska 72a, 40-065 Katowice, Poland; m.michalczyk@awf.katowice.pl (M.M.M.); g.zydek@awf.katowice.pl (G.Z.); j.langfort@awf.katowice.pl (J.L.); 2Department of Sports Training, The Jerzy Kukuczka Academy of Physical Education in Katowice, Mikolowska 72a, 40-065 Katowice, Poland; a.zajac@awf.katowice.pl; 3Department of Methodology and Statistics, The Jerzy Kukuczka Academy of Physical Education in Katowice, Mikolowska 72a, 40-065 Katowice, Poland; a.maszczyk@awf.katowice.pl

**Keywords:** athletes, dietary intervention, anaerobic power, work capacity

## Abstract

Despite increasing interest among athletes and scientists on the influence of different dietary interventions on sport performance, the association between a low-carbohydrate, high-fat diet and anaerobic capacity has not been studied extensively. The aim of this study was to evaluate the effects of a low-carbohydrate diet (LCD) followed by seven days of carbohydrate loading (Carbo-L) on anaerobic performance in male basketball players. Fifteen competitive basketball players took part in the experiment. They performed the Wingate test on three occasions: after the conventional diet (CD), following 4 weeks of the LCD, and after the weekly Carbo-L, to evaluate changes in peak power (PP), total work (TW), time to peak power (TTP), blood lactate concentration (LA), blood pH, and bicarbonate (HCO_3_^−^). Additionally, the concentrations of testosterone, growth hormone, cortisol, and insulin were measured after each dietary intervention. The low-carbohydrate diet procedure significantly decreased total work, resting values of pH, and blood lactate concentration. After the low-carbohydrate diet, testosterone and growth hormone concentrations increased, while the level of insulin decreased. After the Carbo-L, total work, resting values of pH, bicarbonate, and lactate increased significantly compared with the results obtained after the low-carbohydrate diet. Significant differences after the low-carbohydrate diet and Carbo-L procedures, in values of blood lactate concentration, pH, and bicarbonate, between baseline and post exercise values were also observed. Four weeks of the low-carbohydrate diet decreased total work capacity, which returned to baseline values after the carbohydrate loading procedure. Moreover, neither the low-carbohydrate feeding nor carbohydrate loading affected peak power.

## 1. Introduction

Dietary manipulations are an integral part of an athletes training process, related significantly to optimal performance. The mechanisms responsible for improved exercise performance are best recognized for high-carbohydrate diets (HCHO-D), and are attributed to maximizing muscle glycogen content and thereby its availability and utilization during exercise [1,2]. It has been evidenced that a HCHO-D improves performance in both prolonged and low-to-moderate intensity as well as short, high-intensity exercises [3,4,5]. Interestingly, the effect of exercise on markers of mitochondrial biogenesis, expressed by an increase of citrate synthase suscinate dehydrogenase and β-hydroxyacyl-CoA dehydrogenase activities as well as cytochrome c oxidase IV total protein content, a crucial factor for improving aerobic performance, has been observed when carbohydrate restricted diets were utilized [6]. The molecular mechanism that leads to mitochondrial biogenesis is attributed to activation of peroxisome proliferator-activated receptor-γ coactivator-1 (PGC-1α), recognized as the master regulator of mitochondrial gene expression [7]. Thus, dietary modifications such as carbohydrate restriction have emerged as an alternative strategy for athletes to help improve their performance.

Among carbohydrate-restricted diets, high-fat diets (HFD) have been considered as a possible nourishment manipulation to support progress in athlete’s endurance since Phinney et al. [8] showed that duration time of moderate intensity exercise (60% VO_2max_) was maintained in highly trained cyclists being fed a high-fat ketogenic diet (KD). Interestingly, individual data from this study indicated that some athletes consuming the KD were able to extend the time to exhaustion up to 155%, which suggested the occurrence of individuals more or less vulnerable to the HFD. Considering the moderate intensity of exercise applied, the ergogenic effect of the dietary intervention may be explained by higher rates of muscle ketone bodies/fat utilization in relation to carbohydrate metabolism during the aerobic exercise protocol [9,10,11]. Recently Zajac et al. [12] reported a significant increase in VO_2max_ and improved lactate threshold in off-road cyclists after a HFD as a result of favorable changes in body mass and body composition. There are also some studies showing improvements or no detrimental effects of an HFD on some markers of aerobic and anaerobic exercise performance [13,14,15,16]. However, despite several promising results in favor of improved performance during exercise of moderate intensity, there is a lack of ambiguous evidence to support the benefits of HFD in anaerobic endurance in athletes [14,17]. There are also studies that revealed a negative impact of an HFD on exercise performance in healthy sedentary adults [18,19]. Thus, HFD potentially compromises physical performance during high-intensity exercises and thereby may fail implementation of a training program based on such exercises.

Data on the effects of HFD on high-intensity exercise performance, including strength and speed endurance exercise capacity, are less recognized. A very recent study by Paoli et al. [20] showed a reduction in fat mass, with no changes in both muscle mass and strength in gymnasts who followed a HFD for 30 days. One of the first studies exploring this issue revealed that a 3-day HFD decreased the mean power output without changing peak power values in sedentary young men [21]. In contrast to the proposed benefits of fat metabolism for physical performance of moderate exercise, another study showed increased fat oxidation but impaired high-intensity work output [19,22,23]. In case of high intensity exercise (>75% VO_2max_) fat oxidation is gradually suppressed, and replaced by accelerated glycolysis via the pyruvate dehydrogenase (PDH) reaction [24,25]. This phenomenon has been described as the crossover concept [26]. Moreover, during high-intensity exercise, fat metabolism shifts into higher utilization of its intramuscular deposits with a profound reduction of plasma non-esterified fatty acid metabolism [5,27]. 

In the last few decades or so, athletes and scientists have experimented extensively with different dietary procedures to improve body composition, muscular strength and power, as well as overall work capacity [28,29]. Recent studies have focused on a novel concept called “train low” to prepare athletes for competition [30]. This strategy is based on alternative application of low glycogen stores followed by high CHO availability in the training process. This paradigm, which is more frequently applied as the “sleep low” method, assumes beneficial effects of low glycogen availability on mitochondrial biogenesis and aerobic substrate metabolism [31,32], whereas CHO loading allows for high intensity exercise to be performed.

The organization of the training process is a complex task which is normally based on periodization approaches [33]. In competitive sports, the use of short training periods (up to 4 weeks) focused on improving a particular motor ability, while maintaining the level of other abilities, has been named block periodization [34]. A poorly recognized strategy in the training process includes the application of dietary manipulations implemented into consecutive micro-cycles. Considering the above issue, we attempted to verify if a 4-week low-carbohydrate diet (LCD) diet followed by 7 days of HCHO-D affected anaerobic performance measured by means of an all-out lower limb Wingate test. The LCD was implemented into four consecutive micro-cycles in the precompetitive period, followed by a weekly tapering period, in which sport-specific training was combined with a HCHO-D. For the purpose of our study we chose competitive basketball players because there is significant empirical evidence supporting the view that basketball is a sports discipline that requires the performance of multiple dynamic activities relying on both aerobic and anaerobic metabolism. Such an approach is geared at improving tolerance to high intensity workloads in the competitive period of the annual training cycle. 

## 2. Material and Methods

### 2.1. Participants

Fifteen apparently well-trained male basketball players were enrolled in the study. Their basic somatic characteristics expressed as mean values ± SD were as follows: age 23.5 ± 2.2 years; height 194.3 ± 6.4 cm; body mass 92.18 ± 5.1 kg; body mass index (BMI) 24.98 ± 1.86 kg/m^2^. All study participants had at least five years of training experience and competed at the division I level of the Polish Basketball League. Basic somatic characteristics of the study participants are presented in Table 1. During the five weeks of the experiment the athletes were fed a low-carbohydrate diet (LCD) for 4 weeks, followed by 7 days (1 week) of carbohydrate loading (Carbo-L) (Figure 1). There was no washout period between the two feeding procedures. One month before the experiment began all participants consumed a standard conventional diet (CD) (Table 2). Participants were informed of the nature of the investigation and written informed consent was obtained prior to study commencement. All participants were free from any diseases and were not taking medications nor dietary supplements during the study. None of the athletes had previous experience with the LCD and carbohydrate-loading procedures. The experimental protocol was approved by the Ethics Committee of the Jerzy Kukuczka Academy of Physical Education in Katowice, Poland (ethics reference KB-5/2015) and conformed to the principles of the Declaration of Helsinki.

### 2.2. Experimental Design

#### 2.2.1. Dietary Guidelines—Monitoring of Nutritional Intake

The dietary intervention lasted 5 weeks. Before constructing individual isocaloric LCD, and Carbo-L diets, the resting metabolic rate (RMR) and the total daily energy expenditure (TDEE) associated with training were estimated. The TDEE was calculated according to the commonly accepted model (TDEE = AF (Activity Factor) × RMR (Resting Metabolic Rate) [35]. The RMR was measured at the beginning of the experiment, before the 4-week LCD, as well as before the 7-day Carb-L procedure, by means of an ergo spirometer MetaLyzer 3B (Cortex, Leipzig, Germany). AF was determined based on available indicators for athletes 2.0 (high activity) [35]. Also, before the experiment, the subjects were asked to complete the 72-h food diary (two weekdays and one weekend day). The dietary records were estimated by a nutritionist to assess previous feeding habits and daily calorie consumption. The composition of particular diets is presented in Table 2. Before the experiment all participants consumed an isocaloric conventional diet (CD) (Table 2). The CD was composed of 55% carbohydrates, 15% protein, and 30% fat. During the 5 weeks of the experiment the study participants lived in the dormitory and were fed at the academy cafeteria. The meals were prepared in the form of 24-h menus for seven days of the week. All meals were planed and supervised by a nutritionist. The quality and quantity of the food products was strictly controlled, maintaining proper proportions between the major macronutrients. During the five weeks of the experiment the athletes were fed a low-carbohydrate diet (LCD) for 4 weeks, followed by 7 days (1 week) of carbohydrate loading (Carbo-L) (Figure 1). There was no washout period between the two feeding procedures. One month before the experiment began all participants consumed a standard conventional diet (CD). The LCD was composed in such way that from all consumed fats, unsaturated fatty acids (mono and polyunsaturated) constituted 80% of the daily calorie intake. The LCD consisted of 10% carbohydrates, 31% proteins, and 59% fat. In the LCD, the subjects consumed healthy fats, mainly monounsaturated fatty acids from olive oil, dairy products, and nuts, which accounted for more than 50% of all fatty acids consumed. The LCD also contained polyunsaturated fatty acids *n*-6 and *n*-3, in a ratio not exceeding 4–5:1. The diet included the consumption of fish, like mackerel and sardines, which are rich in *n*-3 fatty acids. Additionally, the LCD included high-quality protein products such as fish, meat, eggs and dairy products. While on the LCD the athletes consumed: poultry, fish, beef, veal and lamb, dried beef, chopped meat tartare, carpaccio and cured ham; olive oil, butter, green vegetables without restriction (raw and cooked), boiled eggs, and seasoned cheese (e.g., mozzarella, halloumi). Warm drinks were restricted to tea and coffee without sugar and herbal extracts. The foods and drinks that athletes avoided included alcohol and any sweets like sugar or honey. They also did not consume white bread, pasta, white rice, sweet milk, fruit yogurt, sweets, soluble tea, and barley coffee. During the 4 weeks of the LCD the athletes consumed four main meals and one snack.

The Carbo-L consisted of 75% carbohydrates, 16% proteins and 9% fat. The Carbo-L diet contained carbohydrates, mainly with a low glycaemic index like whole grain bread and pasta, graham rolls, whole grain rice, legumes, raw vegetables, poultry, beef, pork and fish. Only after training, the athletes consumed medium or high glycaemic index snacks like bananas, honey, figs, dactyls or meals with white rice, potatoes, boiled carrots, and beetroots. In the Carbo-L diet, the subjects did not eat processed carbohydrates (fast foods), sweets and carbonated drinks. In the Carbo-L protocol, the participants ate healthy carbohydrates such as cereals, rice, buckwheat, millet, and fruits. They also consumed high-quality protein and fat products, similar to the LCD. Participants consumed four main meals and two snacks. The main meals were prepared and consumed at the cafeteria, while the snacks were packed and eaten after the training sessions. 

#### 2.2.2. Training Program

During the 5 weeks of the experiment three series of laboratory analyses were performed. Baseline evaluations (after CD diet), after four weeks of the LCD, as well as after the 7 day Carbo-L procedure. The study was conducted during the precompetitive period of the annual training cycle. The basketball players performed five training units per week with a scrimmage game played on Saturday. Each training session lasted from 90 to 120 min and included specific technical and tactical drills as well as conditioning exercises. The training intensity varied significantly from low (stretching, free throws—HR ≤ 120 bts/min) to submaximal during transition or full court press drills (HR ≥ 170 bts/min). The participants refrained from exercise for 2 days before testing to minimize the effects of fatigue. 

### 2.3. Body Mass, Body Composition

The subjects underwent medical examinations and somatic measurements. Body composition was evaluated in the morning, between 08.00 and 08.30 h. The day before, the participants had the last meal at 20.00 h. They reported to the laboratory after an overnight fast, refraining from exercise for 48 h. The measurements of body mass were performed on a medical scale with a precision of 0.1 kg. Body composition was evaluated using the electrical impedance technique (Inbody 720, Biospace Co., Japan). 

### 2.4. Anaerobic Performance

Anaerobic performance was evaluated by the 30-s Wingate test for lower limbs. The test was preceded by a 5 min warm-up with a resistance of 100 W and cadence within 70–80 rpm. Following the warm-up, the test trial started, in which the objective was to reach the highest cadence in the shortest possible time, and to maintain it throughout the test. The lower limb Wingate protocol was performed on an Excalibur Sport ergocycle with a resistance of 0.8 Nm·kg^−1^ (Lode BV, Groningen, The Netherlands). The recorded variables included: time to peak power (TTP (s)), peak power (PP (W/kg)), and total work performed (TW (J/kg)), (Lode Ergometer Manager—LEM, software package, Groningen, The Netherlands). 

### 2.5. Biochemical Analysis

To determine lactate concentration (LA) and acid–base equilibrium, the following variables were evaluated: LA (mmol/L), blood pH, bicarbonate (mmol/L). The measurements were performed on fingertip capillary blood samples at rest and after 3 min of recovery. Determination of LA was based on an enzymatic method (Biosen C-line Clinic, EKF-diagnostic GmbH, Barleben, Germany). The remaining variables were measured using a Blood Gas Analyzer GEM 3500 (GEM Premier 3500, Germany). β-hydroxybutyrate (β-HGB-mmol/L) was measured using Randox UK diagnostic kits (Ranbut). Testosterone (T), growth hormone (GH), and insulin (I) concentrations were measured in duplicate using EDTA plasma and immunoassay kits customized on an automated analyzer (Cobas e411, Roche Diagnostics, Mannheim, Germany). The intra assay coefficient of variation was for C 2.2%, 2.5% for T, for GH 2.3%, and 4.6% for insulin. 

### 2.6. Statistical Analysis

Age, body mass and body composition, as well as biochemical variables were expressed as mean ± SD. Before using the parametric test, the assumption of normality was verified using the Kolmogorov–Smirnov test. A one-way ANOVA and two way ANOVA with repeated measures were used with significance set at *p* < 0.05. When appropriate, a Bonferroni post hoc test was used to compare selected data. The remaining analyses were performed using STATISTICA (StatSoft, Inc., Tulsa, OK, USA, 2018, version 12).

## 3. Results

Changes in basic somatic variables after particular dietary interventions in basketball players are presented in Table 1. The analysis of variance revealed statistically significant differences between LCD and CD for body mass (BM) and fat mass (FM) and between Carbo-L and LCD for fat-free mass (FFM) values. There was no significant differences between CD and Carbo-L.

The differences between the CD, the LCD and the Carbo-L diets are shown in Table 2. When comparing the LCD with the CD and the Carbo–L, the LCD contained a substantially higher amount of monounsaturated fatty acids (MUFAs) and polyunsaturated fatty acids (PUFAs) as well as significantly greater amounts of protein. Also, the LCD contained smaller amounts of CHO.

The results of ANOVA for anaerobic variables of the Wingate test are presented in Table 3. The analysis of variance revealed statistically significant differences between CD versus LCD (*p* = 0.002) and between Carbo-L and LCD (*p* = 0.021) for TW values.

The analysis of variance for blood acid–base equilibrium after the LCD revealed statistically significant differences between CD versus LCD and between Carbo-L versus LCD for LA and pH values. The same analysis showed statistically significant differences in values of LA, pH, and HCO_3_^−^ between baseline and post exercise values (Table 4). The same analysis revealed statistically significant differences between rest and post exercise for LA, pH, and HCO_3_^−^ in CD, LCD, and Carbo-L. 

The results of ANOVA for hormone concentrations after particular dietary interventions in basketball players, revealed statistically significant differences between CD versus LCD and between Carbo-L versus LCD for GH (*p* = 0.003) and I (*p* = 0.002) values (Table 5). The same analysis showed significant differences of T concentration between LCD vs CD and between Carbo-L vs CD (*p* = 0.002).

## 4. Discussion

Particular sport disciplines are characterized by different exercise metabolism that requires specific nutrition. Substantial research has been completed on the impact of both carbohydrate and high-fat diets on substrate utilization during exercise, both in sedentary subjects and in athletes. A common observation of these studies is a positive impact on performance during prolonged submaximal exercise [3,36]. However, these ergogenic effects on physical performance are induced by various mechanisms. Chronic ingestion of a high-fat diet results in a greater reliance on fat oxidation at rest and during exercise [37] that allows the muscle to spare glycogen during moderate exercise and therefore improve exercise capacity [38,39]. On the other hand, high-carbohydrate diets elevate muscle glycogen content. This effect can be beneficial not only for endurance performance, but may in fact improve high-intensity exercise, during which muscle glycogen is the predominant fuel source [5]. More recent studies concerning athlete performance and specific dietary interventions have focused on the combined use of high-fat, low-carbohydrate diet, followed by carbohydrate loading [40]. This paradigm assumes that these two different diet interventions can influence exercise glycogen metabolism, which results in favorable effects on performance. We assumed that the glycogen sparing effect of LCD should enhance endurance performance, while the HCHO-D strategy could benefit anaerobic performance after carbohydrate re-feeding. In the present study we utilized this protocol to investigate anaerobic performance in well-trained basketball players with the following new approaches: (1) adaptation to fat metabolism was induced by the LCD diet, (2) a prolonged state of carbohydrate loading (up to one week) was applied, (3) the dietary interventions were implemented into subsequent training micro-cycles, and (4) the 30-s all-out Wingate test was used to evaluate anaerobic power and capacity. In this study, we found that 4 weeks of LCD feeding decreased total work capacity, which returned to baseline after the carbohydrate loading procedure. Moreover, neither the low-carbohydrate feeding nor the carbohydrate loading affected peak power. In case of the LCD diet used in our study, the obtained results are similar to those observed in sedentary, healthy individuals after using a short-term (3-days) ketogenic diet [21]. This means that the training process applied by the basketball players, despite the use of explosive activities and short repeated bursts of high-intensity efforts, did not trigger adaptive changes towards the elimination of negative effects of the LCD on anaerobic capacity, and carbohydrate refeeding did not bring additional benefits related to anaerobic performance. The LCD used by our subjects contained substantially high amounts of MUFAs and PUFAs as well as significant amounts of protein. The high dietary content of the later nutrient was used to protect athletes against protein loss induced by heavy training loads, whereas larger content of PUFAs and MUFAS was consumed to protect against post exercise inflammation and to enhance recovery. Despite the above mentioned modifications, the diet used in our experiment met the criteria of LCD, what was confirmed by four times higher concentrations of β-hydroxybutyrate (β-HB) compared to baseline values.

The results of the all-out Wingate test depend almost exclusively on activation of anaerobic glycolysis [41]. One possible mechanism for reduced anaerobic capacity may be attributed to the LCD-induced keto-acidosis, which was reflected in the present study by reduction of blood pH. It is well established that anaerobic glycolysis is limited by acidosis via inhibition of the rate limiting enzyme, phosphofructokinase [42]. However, the reduced anaerobic capacity may be as well attributed to aerobic re-synthesis of ATP, which depending on the physiological state of the body provides 10–40% of the energy utilized during the 30-s Wingate test. It is known that this metabolic system is less effective during high-intensity exercise after low-carbohydrate diets [43,44]. While a beneficial impact of LCDs on athletes’ performance during low- to moderate-intensity exercise has delivered conflicting results [22], most of the data regarding anaerobic performance indicated its impairment [19,43], although no differences in maximal power output were found when the effects of consuming a high-fat diet were compared with carbohydrate loading, even when both diets were accompanied by intensive training [29].

During the first few seconds of the Wingate test ATP re-synthesis is mainly provided by anaerobic, alactic metabolism, i.e., muscle ad hoc available ATP and breakdown of creatine-phosphate. The later process contributes close to 70% of anaerobic synthesis of ATP and is indirectly expressed by the measurement of peak power. During the current study, in contrast to anaerobic capacity, peak power did not change significantly during the experiment. It is known that alterations in systemic pH strongly affect anaerobic exercise performance [45]. This means that conversely to anaerobic capacity, the LCD which induced subclinical keto-acidosis did not affect this variable. It also indicates that muscle acidosis was fully compensated during the first seconds of the Wingate test by transient muscle alkalization, as a result of creatine-phosphocreatine breakdown. A similar effect, with no changes in peak power, was previously seen in sedentary individuals following a 3-day ketogenic diet [21]. It is well established that high cellular H^+^ concentration is a crucial factor that reduces peak power [46]. These results also suggest that CHO restriction may not be detrimental to perform acute all-out explosive efforts. This result is in agreement with previous data reported by Dipla et al. [47] showing that strength, measured isotonically, isometrically, or isokinetically was maintained during short-term carbohydrate restriction. The above mentioned results suggest that during the first few seconds of an all-out effort, delivery of free energy from ATP breakdown is not impaired by LCD in skeletal muscle metabolism and contracting muscles are able to maintain metabolic stability [48,49].

During this study, macronutrient redistribution towards carbohydrate loading restored total work capacity and blood pH to baseline (pre-experimental) values, yet no benefits appeared in any of the investigated anaerobic variables after carbohydrate re-feeding. These results are consistent with the findings of Burke at al. [14] and Carey et al. [50], who demonstrated an increase in fat oxidation with short-term high-fat feeding that persisted even after restoration of CHO stores. In skeletal muscles, fat feeding promotes utilization of fatty acids and intramuscular triglycerides as a primary fuel source [28,51] by decreasing the activation of pyruvate dehydrogenase complex (PDH) due to increased pyruvate dehydrogenase kinase (PDK4) activity [52], causing fat utilization as a main substrate for ATP re-synthesis. However, this mechanism was perceived only in individuals subjected to aerobic or intermittent exercises after pre-exercise carbohydrate feeding. It is worth mentioning that this type of exercise is predominantly performed with recruitment of slow-twitch fibers. There is still very little information regarding how fast the metabolism of LCD-adapted muscles could be reversed with carbohydrate re-feeding and if metabolic regulation during exercise in fast-twitch fibers which are mostly engaged in execution of all-out efforts is similar to that which takes place in slow-twitch fibers.

The supply of substrate for muscle energy metabolism and its biological effects are regulated by hormones. Research has shown that dietary depletion of CHO can shift hormonal milieu and cellular mechanisms to increased utilization of non-esterified fatty acids (NEFAs) and to a much lesser extent amino acids [53]. The following hormones showing a widespread metabolic effect have been identified as anabolic: T, I, and GH, while C is considered to possess a more potent catabolic property. Hormonal anabolic action is primary mediated by amino acid uptake and transcriptional regulation of selected genes that leads to an increase in intramuscular protein synthesis. This means that elevated levels of these hormones combined with exercise can potentiate skeletal muscle hypertrophy, which plays a significant role in the performance of anaerobic efforts. Such an effect did not occur in our subjects despite the elevation of T levels and relatively low C concentrations, since both the BM and FFM of the tested athletes did not change substantially throughout our study. This could have been caused by the selection of study participants, which included well-trained athletes with a low body fat content and significant amounts of fat free mass due to regular resistance training. Testosterone is also a strong fat-reducing hormone, and exerts this effect by inhibiting lipid uptake and lipoprotein lipase activity in adipocytes, stimulating at the same time lipolysis through increasing the number of lipolytic beta-adrenergic receptors [54]. The latter can elevate blood NEFA levels, thus providing more substrate for ketogenesis. Our study indicates that this effect cannot be considered as a primary source for stimulation of ketogenesis in our subjects because the elevated T level was stable throughout the dietary intervention. In favor of such an assumption is the fact that despite elevated T levels, the concentration of ketone bodies (β-HB) returned to baseline values after the CHO re-feeding. Human studies have demonstrated strong positive relationships between dietary saturated fat and T, which can be considered as a predictor of plasma T level [55]. The fact the increases in T concentrations in our athletes were similar after both dietary interventions indicates that despite increased fat consumption, other exercise-induced factors may play an important role in triggering T production. In one of his studies Volek et al. [55] observed that none of the fat-enriched dietary variables were significantly correlated with C concentrations. A similar effect is confirmed by the present study which showed a lack of changes in plasma C levels after the adherence to both diets. The fact that T and C responded differently in our experimental paradigm suggests involvement of numerous mechanisms for these two hormonal responses. 

As has been pointed out by other authors [53,56], the substantial impact of accelerating the rate of ketogenesis may be a consequence of insulin deficiency. This leads to metabolic changes towards ketone body formation [57]. In line with such an assumption, in the present study ketoacidosis was completely reversed after carbohydrate re-feeding, once the insulin concentration increased above baseline level. The observed ketogenesis in this study was accompanied by significant and parallel changes in insulin concentration, which indicates superior influence of this hormone in regulation of metabolism during the LCD in comparison to other investigated hormones. However, the question also arises as to whether elevated testosterone levels are able to compensate skeletal muscle amino acids uptake during the LCD with decreased insulin concentration. Changes in the GH also contribute to maintenance of metabolic rate in muscles. Its physiological function in keto-adapted individuals has been poorly investigated. It is known that for both endurance and resistance exercise greater activation of anaerobic glycolysis and lactate formation increases the amount of GH released. Similarly to T, GH seems to enhances lipolysis during exercise, while sparing protein as a metabolic source, and causes an increase in muscle mass and strength [58]. 

## 5. Conclusions

Until now, there was no evidence how chronic fat adaptation followed by carbohydrate loading compromises all-out anaerobic exercise metabolism and performance when such a dietary strategy is implemented into the training process. The recommendations for such dietary interventions rely to a great extent on findings from endurance exercise. Results of the present study suggest that supplying the body with an alternative fuel through nutritional manipulation, followed by CHO re-feeding, may not bring additional benefits to athletes involved in sport disciplines with a predominance of anaerobic metabolism. Thus, our findings call into question such dietary interventions when optimizing all-out intense anaerobic exercise performance. Several studies have focused on the potential role of KD interventions on athletic performance, which could be used by coaches in parallel with training. In various sports that rely on anaerobic performance (i.e., wrestling, martial arts, boxing, gymnastics, etc.), athletes are often required to reduce body mass in a short period of time. Rapid weight loss via reduced food intake, sweat suits, diet pills, and dehydration methods is often associated with loss of an optimal physical and mental fitness. This problem can be partially solved by implementation of the LCD into the training program. Although it is very likely that such nutritional manipulation may reduce anaerobic performance, the results of our study indicate that this detrimental side-effect can be reversed by re-feeding athletes with a HCHD. However, such nutrition interventions should be applied several weeks before competition. 

## Figures and Tables

**Figure 1 nutrients-11-00778-f001:**
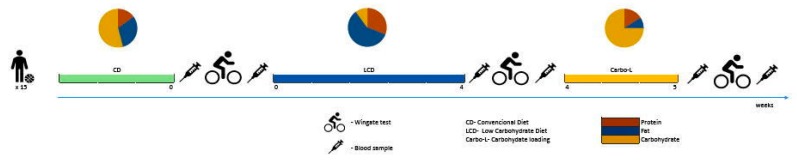
Scheme of the experimental protocol.

**Table 1 nutrients-11-00778-t001:** Body mass and body composition results after CD, LCD, and Carbo-L diets.

Variables	After CD	After LCD	After Carbo-L
	Mean ± SD	Mean ± SD	Mean ± SD
BM (kg)	92.18 ± 5.17	90.38 ± 3.12 *	91.82 ± 4.32
BMI (kg/m^2^)	24.48 ± 0.18	23.93 ± 0.17	24.19 ± 0.21
FFM (kg)	79.62 ± 4.88	78.20 ± 3.65	79.92 ± 3.84 #
FM (%)	12.42 ± 2.25	11.1 ± 1.25 *	11.8 ± 1.23

Note: BM—body mass, BMI—body mass index, FFM—fat-free mass, FM—fat mass, LCD—low-carbohydrate diet, CD—conventional diet, Carbo-L—carbohydrate loading. * *p* < 0.05 significant difference to the after CD (*p* < 0.05), # statistically significant difference (*p* < 0.05) Carbo-L vs. LCD.

**Table 2 nutrients-11-00778-t002:** Average macronutrients and total energy intake during the CD, LCD, and Carbo–L dietary procedures.

Nutrients	CD	LCD	Carbo-L
	Mean ± SD	Mean ± SD	Mean ± SD
Carbohydrates (%)	54 ± 6.1	10 ± 0.5	75 ± 3
Protein (%)	15 ± 6.3	31 ± 2.3	16 ± 3
Fat (%)	31 ± 4.3	59 ± 3.6	9 ± 1.6
SFAs (g)	48 ± 6.1	30 ± 4.2	11 ± 2.4
MUFAs (g)	61 ± 5.2	128 ± 12.3	13 ± 1.6
PUFAs (g)	20 ± 2	68 ± 4.5	10 ± 1.7
*n*-3 (g)	3.2 ± 0.2	24.4 ± 0.6	1.8 ± 0.3
*n*-6 (g)	16.1 ± 6	47.7 ± 2.7	7.5 ± 1.2
*n*-6/*n*-3	5 ± 1	2 ± 1	4 ± 1
TEI (kcal)	3740 ± 53	3758 ± 42	3752 ± 15
TEI (kJ)	15,658.63 ± 221	15,733.99 ± 175	15,708.87 ± 62

Note: SFAs—saturated fatty acids, MUFAs—monounsaturated fatty acids, PUFAs—polyunsaturated fatty acids, *n*-3—omega 3, *n*-6—omega 6, TEI—total energy intake.

**Table 3 nutrients-11-00778-t003:** Anaerobic variables of the Wingate test after particular dietary interventions in basketball players.

Variables	CD	LCD	Carbo-L
	Mean ± SD	Mean ± SD	Mean ± SD
TPP (s)	2.65 ± 0.61	2.73 ± 0.57	2.58 ± 0.39
PP (W/kg)	20.35 ± 3.44	19.94 ± 3.42	20.87 ± 0.39
TW/kg (J/kg)	301.17 ± 12.42	**266.69 ± 6.46** *	**302.46 ± 8.50** #

Note: TPP—time to peak power; PP—peak power; TW—total work; * statistically significant difference LCD vs. CD; # statistically significant difference Carbo-L vs. LCD.

**Table 4 nutrients-11-00778-t004:** The differences in the concentration of blood plasma lactate and acid-base variables, as well as the resting concentration of β-HB after particular dietary interventions in basketball players.

Variables		CD	LCD	Carbo-L
		Mean ± SD	Mean ± SD	Mean ± SD
LA (mmol/L)	Rest	1.65 ± 0.06	1.26 ± 0.01 *	1.69 ± 0.04 #
	Post exercise	9.47 ± 1.04 ^&^	8.36 ± 0.62 ^&^	9.62 ± 0.54 ^&^
pH (−Log[H^+^])	Rest	7.412 ± 0.003	7.381 ± 0.001 *	7.420 ± 0.01 #
	Post exercise	7.275 ± 0.005 ^&^	7.322 ± 0.03 ^&^	7.261 ± 0.008 ^&^
HCO_3_^−^ (mmol/L)	Rest	24.10 ± 0.07	23.72 ± 0.11	24.48 ± 0.07
	Post exercise	12.80 ± 0.09 ^&^	13.12 ± 0.14 ^&^	12.7 ± 0.05 ^&^
β-HB (mmol/L)	Rest	0.041 ± 0.02 *	0.161 ± 0.11	0.035 ± 0.01 #

Note: LA—lactate; β-HB—β-hydroxybutyrate, HCO_3_^−^—bicarbonate ***** statistically significant differences with *p* < 0.05 between LCD vs. CD; # statistically significant differences with *p* < 0.05 between Carbo-L vs. LCD; ^&^ statistically significant differences with *p* < 0.05 between rest vs. post exercise.

**Table 5 nutrients-11-00778-t005:** Differences in hormone concentrations after particular dietary interventions in basketball players.

Variables	CD	LCD	Carbo-L
	Mean ± SD	Mean ± SD	Mean ± SD
Testosterone (nmol/L)	546.67 ± 167.16	642.37 ± 194.47 *	643.14 ± 186.52 ^$^
Growth hormone (ng/mL)	0.15 ± 0.07	0.21 ± 0.09 *	0.11 ± 0.08 #
Insulin (IU/mL)	5.49 ± 3.25	3.99 ± 2.61 *	7.28 ± 3.65 #
Cortisol (µg/dL)	16.38 ± 6.81	16.22 ± 6.40	16.02 ± 5.79

Note: * statistically significant differences with *p* < 0.05 between LCD vs. CD; # statistically significant differences with *p* < 0.05 between Carbo-L vs. LCD; ^$^ statistically significant differences with *p* < 0.05 between Carbo-L vs. CD.

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
