# Peer review of "Anaerobic Performance after a Low-Carbohydrate Diet (LCD) Followed by 7 Days of Carbohydrate Loading in Male Basketball Players"

_nutrients, 2019, doi:10.3390/nu11040778_

Round 1

Reviewer 1 Report

Short summary

The aim of this paper is to provides an understanding on the effects of a 4 weeks low carbohydrate diet (LCD) followed by seven days of carbohydrate loading (Carbo-L) on time to pick power, peak power total work performance. The study was carried out on 15 male Basketball players. Results suggest that a diet low carbohydrate and high fat diet, followed by Carbohydrate rich diet combined with activities relying on both aerobic and anaerobic metabolism may increase Testosterone levels. Total work decreased significantly after the LCD but increased after the CARB-L.

The Title

(1)     Since it seems that only male basketball players took part in the study, the title should read “…in male basketball players”, since sex-specific differences are not unlikely.

The Abstract

The Abstract appropriately summarize the manuscript.

(2)     However, I encourage the authors, if possible, to include some background

(3)     Row 16: (6)          As well as in the title the following sentence should read: fifteen competitive male basketball players …

(4)      Row 25: I find this sentence “Four weeks of an LCD decreased total work capacity, which returned to baseline values after the carbohydrate loading procedure. Moreover, neither the low carbohydrate feeding nor carbohydrate loading affected peak power” misleading.

(5)     According to Table 3 total work capacity decreased significantly after the LCD and increased after the significantly (LCD vs CARBO-L) after the Carbo-L. However, there was no analysis of CD vs Carbo_L. Although the Values seem similar I recommend the authors to prove it statistically 

The Introduction

(6)     The authors clearly state in the introduction the specific objective of the research and provide the rationale of the study. I find this section very well structured. The first paragraph mentions the issue that summarizes the background addressing the nature of the research question.

(7)     The aim of the study is clear and concisely defined 

The Methods Section

In this section I encourage the authors to include the points mentioned below in order to avoid concerns of potential bias and provide future investigator a complete blueprint to reproduce the study or differences in the outcome of similar studies.

(8)     Participants

(i)                   Mean age

(ii)                 Mean BMI

(iii)                Could you further describe the body competition of participants perhaps in table 1

(9)      Row 97-100: I find this description more appropriated for point 2.3 3. Experimental design and training program.

(10) I think also this point, Experimental design, should be after description of participants and divided into dietary intervention and training program to give a clear overview of the study design.

(11) Dietary guidelines – monitoring of nutritional intake

(i)                   Table 2 displays results of nutrients intake, therefore, should be placed in the results or supplement section.

(ii)                 Here I would also encourage the authors to divide the dietary intervention section in “thinks that were similar in both diets”, LCD and Carbo-L giving in each a full description of the respecting diet avoiding comparisons between diets in order to make it clearer and friendlier for the riders. For instance:

Energy intake as well as macro and micronutrient intake of all subjects were determined by the 24 h nutrition recall. All participants consumed an isocaloric standard Conventional Diet (CD. The CD was composed of 55% carbohydrates, 15% protein and 30% fat. All meals were planed and supervised by a nutritionist. The quality and quantity of the food products was strictly controlled, maintaining proper proportions between the major macronutrients. During the five weeks of the experiment the athletes were fed a Low Carbohydrate Diet (LCD) for 4 weeks, followed by 7 days (1 week) of carbohydrate loading (Carbo-L) (Figure 1 IS MISSING). There was no washout period between the two feeding procedures. One month before the experiment began all participants consumed a standard Conventional Diet (CD).

-          LCD

The LCD was composed in such way that from all consumed……

In the LCD, the subjects consumed healthy fats, mainly monounsaturated fatty acids from olive oil…..

During the 4 weeks of the LCD the athletes consumed 4 main meals and 1 snack. Carbo-L

The Carbo-L diet contained….

the Carbo-L diet, the subjects did not eat processed carbohydrates (fast foods), sweets….

Participants consumed 4 main meals and 2 snacks…

(iii)                 Row 127-128:The meals were prepared in the form of 24h menus for seven days of the week.” It seems that meals were prepared in the form of 24h menus for seven days of the week just in the Carbo-L diet. If not, this phrase should be pleased in the upper section

(12) Experimental design and training program

(i)                   In the introduction authors mention that the LCD was implemented into four consecutive micro-cycles in the precompetitive period. Since I’m not an expert on the field, I was hoping to find more information about this micro-cycle in this section.

Statistical analysis

(13) To the best of my knowledge the statistical methods chosen by the authors are appropriated for the analysis.

(14) In my opinion analysis CD vs Carbo-L of most variables is missing (table3 and 4) to assess the effects of a low carbohydrate diet/ high fat diet, followed by Carbs re feeding.

The Results

(15)  In my opinion authors should include here Table 2 as a compliance analysis according to the 24 hr. recalls, and Table 1 as results of body composition.

(16) If Authors decide that tables are fine in the method section, a description of anthropometrical results would be interesting in the results section

(17)  Have you analyzed statistically CD vs Carbo-L for variables in table 3 and 4?

The Discussion Section

Figures and Graphs

Table 1

I recommend the Authors to have Tables uniformity if possible

(i)                   Include BMI

(ii)                 According to the legend there is no significant difference after the CD. If so, I encourage the Authors to include P values in the Table similar than Table 3 to achieve uniformity 

(iii)                Table 4 and 5 check for dots instead of commas

(iv)                Table 4: Standard deviations for ß-HB are missing

(v)                 If p values are included on the tables *, #, $ are not necessary. Maybe just highlight in bold the significant values. However, it is unclear what does “& - statistically significant differences with p<0.05 between baseline vs after effort” refers to. CD vs Carbo-L or rest vs post exercise?

(vi)                 Figure 1 is missing

Author Response

Response to reviewer 1

Ad1. We fully agree with the reviewer that a clarification is needed. The new title reads:

Anaerobic performance after a low carbohydrate diet (LCD) followed by 7 days of carbohydrate loading in male basketball players.

Ad2. The authors are fully aware that the background could complement the content, however the restrictions on the number of characters originally limited the length of the text. We have corrected the abstract in accordance to the suggestions of the reviewer.

Ad3. Corrected as suggested.

Ad 4. Four weeks of an LCD decreased total work capacity, which returned to baseline values after the carbohydrate loading procedure. Moreover, neither the low carbohydrate feeding nor the carbohydrate loading procedure affected peak power.

Ad 5. We fully agree with the reviewer that the suggested analysis complements the data and increases the scientific value of the manuscript.

Ad 6,7. The authors thank the reviewer for complementing this part of the manuscript.

Ad 8. We fully agree with the reviewer that some of the somatic data should be included in table 2.

Ad 9,10,11. The authors agree with all suggestions of the reviewer. The text has been changed.  

Ad 12. The male basketball players performed 5 training units per week with a scrimmage game played on Saturday. Each training session lasted from 90 to 120 min and included specific technical and tactical drills as well as conditioning exercises. The training intensity varied significantly from low (stretching, free throws – HR ≤ 120 bts/min) to submaximal during transition or full court press drills (HR ≥ 170 bts/min).

Ad 13. We thank the reviewer for acknowledging the choice of statistical methods and the proper analysis of the obtained results.

Ad 14. We have compared and analyzed statistically all considered variables between the CD and the Carbo-L diet. The results are now included in tables 3 and 4.

Ad 15,16. The authors decided to move table 1 and 2 to the results section.

Ad 17. This issue was addressed in the 14th point of the response to reviewer.

Ad 18. All of the suggestions of the reviewer regarding the uniformatity of tables and figures have been included. The significant values have been marked in bold and the missing SD have been added.

We thank the reviewer for all the comments, which we have addressed in the response hoping that the text has been significantly improved and can now be accepted for print in the special issue of Nutrients. 

Reviewer 2 Report

The authors have performed a well-designed study, as the results of blood lipid changes already are published. Many studies regarding low-carb diets and performance are available in the literature. However, there is still a public interest of new reports. The present study will probably meet its audience, but a major revision is needed regarding the papers layout, also to make it easier for the reviewers to make a proper evaluation of the paper and improve its readability.

Different fonts are used in the same sentences i.e row 35, 36 and 40.

Many abbreviations are used. Many are common used, but many are unnecessary i.e in table 2, Pro could be spelled out protein, and the table text below uses uppercase letters. Not all abbreviations are explained in the text, i.e row 66; PDH reaction, section 2.6 Biochemical analysis, i.e BE. Also, some of these results are lacking in the text, and should be presented in table. Lactate is abbreviated La and LA.

The layout of the tables must be improved, See table 1 and 2.

The title focus on anaerobic performance, and therefore the interesting data in table 3 should get more focus, maybe in a graph.

To be able to make a proper review of the present paper, it would be valuable to have an improved text, with a homogenous layout regarding the text mass, i.e font, size, tables and abbreviations. As in its present format, it really confuses the reader and makes it hard to give fair critics.

Author Response

Response to reviewer 2

Ad 1,5. The major mistakes in the text, which include different fonts, commas and dots has been corrected.

Ad 2. The abbreviations have been explained and unified.

Ad 3. We fully agree with the reviewer that the tables 1 and 2 needed improvement. They have been corrected and supplemented with additional data, especially regarding the differences between the CD and the Carbo-L diet.

Ad 4. The data from table 3- anaerobic variables, has been exposed.

We thank the reviewer for all the comments, which we have addressed in the response hoping that the text has been significantly improved and can now be accepted for print in the special issue of Nutrients. 

Round 2

Reviewer 2 Report

The authors have really improved the quality of the manuscript, and I have no more comments to add.

Author Response

We would like to thank the reviewer for careful and thorough reading of this manuscript and for the thoughtful comments and constructive suggestions, which help to improve the quality of this manuscript.